# In Silico Analysis of SARS-CoV-2 Spike Proteins of Different Field Variants

**DOI:** 10.3390/vaccines11040736

**Published:** 2023-03-27

**Authors:** Muhammad Haseeb, Afreenish Amir, Aamer Ikram

**Affiliations:** Department of Microbiology, National Institute of Health, Islamabad 45500, Pakistan

**Keywords:** coronaviruses, SARS-CoV-2, Alpha variant (B.1.1.7), Delta variant (B.1.617.2), GISAID, in silico analysis

## Abstract

Coronaviruses belong to the group of RNA family of viruses that trigger diseases in birds, humans, and mammals, which can cause respiratory tract infections. The COVID-19 pandemic has badly affected every part of the world. Our study aimed to explore the genome of SARS-CoV-2, followed by in silico analysis of its proteins. Different nucleotide and protein variants of SARS-CoV-2 were retrieved from NCBI. Contigs and consensus sequences were developed to identify these variants using SnapGene. Data of the variants that significantly differed from each other was run through Predict Protein software to understand the changes produced in the protein structure. The SOPMA web server was used to predict the secondary structure of the proteins. Tertiary structure details of the selected proteins were analyzed using the web server SWISS-MODEL. Sequencing results showed numerous single nucleotide polymorphisms in the surface glycoprotein, nucleocapsid, ORF1a, and ORF1ab polyprotein while the envelope, membrane, ORF3a, ORF6, ORF7a, ORF8, and ORF10 genes had no or few SNPs. Contigs were used to identify variations in the Alpha and Delta variants of SARS-CoV-2 with the reference strain (Wuhan). Some of the secondary structures of the SARS-CoV-2 proteins were predicted by using Sopma software and were further compared with reference strains of SARS-CoV-2 (Wuhan) proteins. The tertiary structure details of only spike proteins were analyzed through the SWISS-MODEL and Ramachandran plots. Through the Swiss-model, a comparison of the tertiary structure model of the SARS-CoV-2 spike protein of the Alpha and Delta variants was made with the reference strain (Wuhan). Alpha and Delta variants of the SARS-CoV-2 isolates submitted in GISAID from Pakistan with changes in structural and nonstructural proteins were compared with the reference strain, and 3D structure mapping of the spike glycoprotein and mutations in the amino acids were seen. The surprisingly increased rate of SARS-CoV-2 transmission has forced numerous countries to impose a total lockdown due to an unusual occurrence. In this research, we employed in silico computational tools to analyze the SARS-CoV-2 genomes worldwide to detect vital variations in structural proteins and dynamic changes in all SARS-CoV-2 proteins, mainly spike proteins, produced due to many mutations. Our analysis revealed substantial differences in the functionality, immunological, physicochemical, and structural variations in the SARS-CoV-2 isolates. However, the real impact of these SNPs can only be determined further by experiments. Our results can aid in vivo and in vitro experiments in the future.

## 1. Introduction

On 31 December 2019, COVID-19 was initially discovered in Wuhan, China. The condition became severe when many infected cases were reported in the “Huanan Seafood Market” [1]. HKU1, HCoV229E, HCoVOC43, HCoVNL63, and HCoV229E are coronaviruses generally responsible for only minor common cold and respiratory infections in newborn infants and the elderly [2]. Based on genetic material properties, the coronavirinae family contains four genes: Alpha, Beta, Gamma, and Delta coronavirus. Coronaviruses are RNA viruses that can trigger an infection in mammals, humans, and birds. They cause respiratory infections such as the common cold, SARS, and MERS [3]. SARS-CoV-2 is a positive-polarity single-stranded RNA virus. It spread extremely rapidly, and became a worldwide pandemic within a few months. Its transmission between individuals and populations is relatively easy because of its transmission pattern in direct body contact and through respiratory droplets from an infected individual [4]. The virus has a 2 to 14-day incubation period and causes severe respiratory problems. Its symptoms are high-grade fever, non-productive cough, pharyngitis, muscle joint pains, runny nose, diarrhea, and shortness of breath. In certain circumstances, loss of sensations such as taste and smell are lost [4].

The four critical structural proteins found in the virion are the N-Protein (nucleocapsid), M-protein (transmembrane), E-Protein (envelope), and S-Protein (spike). However, the direct assembly of structural proteins is not required to form the whole virion infection in some coronaviruses; other proteins with overlapping compensatory roles may be expressed [5,6,7]. The SARS-CoV-2 genome comprises two enormous linear ORFs: ORF1a and ORF1b ORF1a are transcribed into polyproteins 1a and 1b, produced via a one ribosomal frameshift. For replication and transcription, there are 16 non-structural proteins (NSPS) [8].

However, the worldwide spread of COVID-19 has raised significant concerns about viral evolution and adaptation in terms of how it spreads worldwide, encountering various host immune systems and countermeasures determined by mutations, deletions, and recombination.

SARS-CoV-2 varies from earlier strains by having numerous hazardous residues in the coronavirus receptor-binding region (especially Gln493), which delivers valuable communication with ACE2 human receptors [9]. Understanding surface receptor variations in the field are crucial for developing a stable vaccine strain. Therefore, we employed in silico screening of the whole genome for significantly spiked protein SARS-CoV-2 of coronavirus variants from the published data worldwide to understand the viral variation patterns. As a result, SARS-CoV-2 positive samples were sequenced to investigate the genetic diversity during the fourth wave of the epidemic in Pakistan.

There is a pressing need to combat COVID-19, and we need quick and reliable approaches. SARS-CoV-2 varies from earlier strains by having numerous residues in the coronavirus receptor-binding region, which enables its binding with ACE2 human receptors. The change in closeness perhaps explains why this virus is more transmissible than other viruses. Using in silico techniques, we studied the pattern of variation in the proteins, especially the spike proteins, of various SARS-CoV-2 strains reported across the globe. This research will help lay the foundation for future research.

## 2. Methods

### 2.1. Viral Strain Selection and Retrieval of Protein Sequence

First, the proteome of the SARS-CoV-2 virus was taken from the NCBI GenBank (www.ncbi.nlm.nih.gov), and for further analysis, sequences of amino acids were obtained in the FASTA format. The target proteins consisted of membrane proteins, nucleocapsid phosphoproteins, surface glycoproteins, envelope proteins, and open read fragments including ORF1a polyprotein, ORF1ab polyprotein, ORF3a, ORF6 protein, ORF7a protein, ORF7b protein, ORF8 protein, and ORF10 protein.

### 2.2. Mutations Identification in SARS-CoV-2 Genomes

Contigs were made to identify variations using Artemis (https://svibs.com/artemis-modal/ accessed on 20 March 2022) version 8.4 and SnapGene (https://www.snapgene.com/https://svibs.com/artemis-modal/ accessed on 12 April 2022) version 5.3 software, and consensus sequences were developed.

### 2.3. Protein Secondary Structure

The secondary structure of SARS-CoV-2 proteins was anticipated by using the secondary structure prediction method (SOPMA) (https://npsa-prabi.ibcp.fr/cgi-bin/npsa_automat.pl?page=/NPSA/npsa_sopma.html, accessed on 26 April 2022). By keeping the default parameters, the value of the conformational state was 4 (helix, sheet, turn, and coil); the similarity threshold was 8, window width was 17 (Combet et al., 2000) [10]. PredictProtein (https://predictprotein.org/accessed on 17 May 2022) was used to predict the protein structural and functional features, which has the following components:Secondary structure prediction and solvent accessibility;Topology (TMSEG);Disered region (Meta-Disorder);Protein binding (ProNA);Disulfide bond;Conversation score;Disordered region (Meta-Disorder).

### 2.4. Tertiary Structure

The tertiary structure details of the spike proteins were analyzed by using web online software SWISS-MODEL (https://swissmodel.expasy.org/interactive, accessed on 10 June 2022) and Ramachandran plots (https://swift.cmbi.umcn.nl/servers/html/ramaplot.html, accessed on 10 June 2022).

### 2.5. Comparison of Pakistani Variants with Reference Strain

The GISAID database was used to retrieve genome sequences from Pakistan. All relevant sequences from the search results were retrieved in FASTA format (accessed 1 November 2021). CoVsurver was used, which was authorized by GISAID, to analyze our sequences in the FASTA format. These tools are online-based bioinformatics software that have been validated to identify and reassemble new coronavirus isolates. In comparison to SARS-CoV-2, we discovered nucleotide and amino acid mutations as well as correlations. The GISAID CoVsurver software was used to identify the GISAID clade of the sequences. The CoVsurver tool conducts sequence alignments and annotations as well as 3D structure mapping and mutations in amino acids.

## 3. Results

### 3.1. Mutations Identified in the Sequenced SARS-CoV-2 Genomes

The sequencing results demonstrated numerous single nucleotide polymorphisms (SNPs) in surface glycoproteins, nucleocapsids, ORF1a polyprotein 1ab (ORF1ab), and ORF1ab polyproteins. These mutation hotspots may be particularly important in adapting SARS-CoV-2s to the human host. The envelope, membrane, ORF3a, ORF6, ORF7a, ORF8, and ORF10 genes had no or few SNPs, so limited alterations in these proteins might mean that they have conserved activities that are required for viral transmission.

Contigs were made for the identification of variations in the Alpha variant (B.1.1.7) and Delta variant (B.1.617.2) of SARS-CoV-2 with a reference strain (Wuhan) by using SnapGene (https://www.snapgene.com/, accessed on 12 February 2022) version 5.3 software, and consensus sequences were developed as shown in the Figure 1 and other are shown in Appendix A.

### 3.2. Protein Secondary Structure

The secondary structures of the envelope protein, membrane glycoprotein, nucleocapsid phosphoprotein, ORF10 protein, ORF1a polyprotein, ORF1ab polyprotein, ORF3a protein, ORF6 protein, ORF7a protein, ORF7b protein, ORF8 protein, and surface glycoprotein of SARS-CoV-2 were predicted by using the Sopma secondary structure prediction method https://npsa-prabi.ibcp.fr/cgi-bin/npsa_automat.pl?page=/NPSA/npsa_sopma.html, accessed on 26 April 2022. By keeping the default parameters, several conformational states: 4 (helix, sheet, turn, coil); similarity threshold: 8; window width: 17 were further compared with the reference strain of SARS-CoV-2 (Wuhan) proteins, which all showed similarity and variance to each other, as shown in Table 1 [10]. PredictProtein (https://predictprotein.org/ accessed on 17 May 2022) was used to predict the protein structural and functional features. It was only used for those proteins with a drastic change in the secondary structure, as shown in Figure 2 and others are shown in Appendix A.

### 3.3. Tertiary Structure

Tertiary structural details of the spike proteins were analyzed using the online software SWISSMODEL (https://swissmodel.expasy.org/interactive, accessed on 10 June 2022) and Ramachandran plots (https://swift.cmbi.umcn.nl/servers/html/ramaplot.html, accessed on 10 June 2022). Through the Swiss model, a comparison of the tertiary structure model of the SARS-CoV-2 spike protein of the Alpha and Delta variants was made with the reference strain (Wuhan), as shown in Figure 3 and Figure 4 [11].

### 3.4. Comparison of Pakistani Variants with Reference Strain

The Alpha and Delta variants of the SARS-CoV-2 isolates submitted in GISAID (https://www.epicov.org/epi3/frontend#43f233, accessed on 1 November 2021) from Pakistan with changes in the amino acids in the structural and nonstructural proteins compared with the reference strain (hCoV-19/Wuhan/WIV04/2019), as shown in Appendix A, and the 3D structure mapping of spike glycoprotein and mutations in amino acid are shown in the Figure 5.

## 4. Discussion

In this research, we studied two complete sequences of the Alpha and Delta variants of SARS-CoV-2 with the Wuhan variant as the reference sequence. Several mutations have been noticed in the COVID-19 proteins as new variants emerge. These variations have several adverse effects on the structure and function of COVID-19 proteins, making it challenging to administer the COVID-19 complex. Such mutations are discussed here.

Coronaviruses have the biggest genome size ranging from 26.4 to 31.7 kb of all RNA viruses [12,13]. Its enormous gene size allows for greater flexibility in integrating and modifying genes [12,13,14]. In RNA viruses, the mutation frequency is relatively high, increasing virulence and developing new species [15]. The greater rate of mutation within the genomes of viruses in various geographical areas is also one reason that COVID-19 is liable for the changes in the death rate and disease symptoms [16]. In COVID-19, we found a few additional single amino acid changes in the Alpha and Delta variants compared to the reference strain (Wuhan), as shown in the above figures.

Virus particles contain the RNA genetic material and the structural proteins required for host cell entry. Once within the cell, the infecting RNA encodes structural proteins that form viral particles, non-structural proteins that regulate viral assembly, transcription, replication, and control of the host cell, and accessory proteins whose role is unknown. The large genome, ORF1ab, has overlapping open reading frames that encode the polyproteins PP1ab and PP1a. The polyproteins are degraded into 16 non-structural proteins known as NSP1-16. A-1 ribosomal frameshifting occurrence is required to produce the longer (PP1ab) or shorter (PP1a) protein. Based on similarities to other coronaviruses, the proteins include the papain-like protein (NSP3), 3C-like proteinase protein (NSP5), RNA-dependent RNA polymerase (NSP12, RdRp), helices (NSP13, HEL), endoRNAse (NSP15), 2′-O-ribose-methyltransferase (NSP16), and other non-SARS-CoV-2 non-structural proteins and have the functions of viral transcription, replication, proteolytic processing, host immune response suppression, and host gene expression suppression.

Previous research has revealed that mutations in non-structural proteins 2 and 3 play a crucial role in infectious capacity and are mainly accountable for the SARS-CoV-2 differentiation process [16], and that the coronavirus nucleocapsid protein is required for RNA replication, genome packing, and transcription [17]. In addition, the envelope protein is involved in the viral genome assembly and development of ion channels (IC), which are critical for virus-host connection and are primarily related to pathogenesis [5,18].

The spike protein is about 180 to 200 kDa and has 1273 amino acids [19]. To avoid the host’s immunological reaction, many polysaccharide molecules cover the spike protein’s surface [20]. The RBD of the spike protein is the area that mainly interacts with ACE2, leading to virus entry into the host cell [16,21]. For several years, theoretical or experimental techniques have been used to predict protein stability [22]. According to prior studies, a single point mutation in RBD disrupts the antigenic structure, affecting RBD binding to ACE2 [23,24]. Furthermore, in silico investigations have demonstrated that point mutations inside the RBD of spike glycoprotein had a stabilizing impact on the spike protein and were discovered to enhance the protein stability.

The potential changes in the spike protein of COVID-19 discovered in the Alpha variants were H69del, V70del (69), Y144del (143), N501Y, A570D, D614G, P681H (674), T716I S982A, and D1118H, and potential changes in the spike protein in the Delta variability were T19R (20), G142D, E156G, F157del, R158del, L452R, T478K, D614G, P681R (674), and D950N.

Some mutations may play a vital role in binding to human ACE2 receptors and many mutations may have enhanced the binding affinity of the surface glycoproteins. Additional beta strands and hydrogen bonds were shown in both the predictions and the 3D models. These extra mutations may produce structural conformational alterations and a greater binding affinity. Biologists and others can utilize these methods to gain a preliminary knowledge of SARS-CoV-2 mutations and their links to SARS-CoV and other related viruses. The initial observations can then be further investigated using various specialized tools and methods. In particular, computational techniques are promising. Machine learning, artificial intelligence, data integration and mining, visualization, computational and mathematical modeling of critical biochemical interactions, and disease control mechanisms can provide cost- and time-efficient solutions.

Researchers are collecting data related to coronavirus to fully understand the spread of the disease, its pathogenesis, and biology to eliminate it [25]. The explosive growth of structural and genomic databases, combined with computational approaches, contributes to discovering and manufacturing novel vaccination candidates. In addition, modern breakthroughs in immunological bioinformatics have resulted in various tools and web servers that can help cut the time and cost of manufacturing traditional vaccinations.

In our findings, forty substantial mutations were seen in the Alpha and ninety-two substantial mutations in the Delta variant in genomic sequences of Pakistani SARS-CoV-2 strains compared with the Wuhan reference strain, covering the whole viral genome. The additional evaluation found that most of these changes were associated with a few viral genomic regions including the spike, nucleocapsid protein, NS3, NSP2, and NSP6, as structural protein integrity is critical for the immune response. Hence, we looked for mutations in the viral proteins. The nucleocapsid protein is an immunogenic phosphoprotein that helps in genome replication and regulation and the cell signaling pathway. The protein structure is disrupted due to the mutation G204R in the nucleocapsid and D614G in the spike protein. Several studies have revealed that the mutation enhances viral infectivity. Moreover, the D614G mutation expands the spike protein, which might result in protein instability and lead to increased viral infection.

We found 29 mutations in the spike protein of the Delta variant and 14 mutations in the Alpha variant in the genomic sequences of the Pakistani SARS-CoV-2 strains compared with the reference strain. The SARS-CoV-2 spike protein is a prominent target for therapeutic and vaccine development due to its interaction in the host cell receptor identification, attachment, and entrance [26,27]. In the spike protein of both the Alpha and Delta variants, we discovered a D614G (aspartic acid to glycine) mutation. A recent study revealed that the D614G variant is more pathogenic, with infected individuals having a higher viral load; however, there was no correlation with disease severity [28]. In silico studies using pseudoviruses by Li et al. revealed that the D614G mutation significantly enhanced the infection [29]. Similar findings were reported, with the D614G mutant having the highest cell entrance among the spike variations [30]. Furthermore, Hou et al. observed that a change in D614G improves the SARS-CoV-2 infection rate and transmission, primarily in humans and animals [31]. The D614G mutation is becoming a more common strain around the world.

The host diversity of coronaviruses and the variation in tissue tropism is primarily because of changes in the surface glycoprotein. The S1 subunit is linked to functions of the receptor binding domain, while the S2 subunit helps in facilitating virus fusion with cell membranes.

In Pakistan, there was a 66% increase in SARS-CoV-2 cases and 64.8% increase in deaths in June 2020 compared to February–May 2020 (72,460 confirmed cases and 1543 deaths) [32]. The significant rise in the incidence cases and the number of deaths may point toward the widespread distribution of the D614G mutation, which must be further examined [33].

Additional molecular epidemiological investigations are required to track the DG614 strain’s circulation in Pakistan, which could also serve to understand the impact of SARS-CoV-2 gene mutations on disease severity. Moreover, tracing variations in the SARS-CoV-2 spike glycoprotein is critical because of its function in cell receptor interaction, entrance in the host cell, and triggering antibody responses, as the widespread distribution of the D614G variation throughout the world has an influence on vaccination effectiveness. This has been a major concern, as most vaccines were designed on the D614G variation. The same issue has been conveyed through the results of Weissman et al., who showed that the D614G mutation is neutralized at a higher level by serum from vaccinated mice, non-human primate, and humans [34]. Furthermore, regular monitoring of SARS-CoV-2 spike protein gene mutation is essential for detecting escape variants and in the future, for vaccine development.

This emphasizes that the SARS-CoV-2 strains circulating in Pakistan have mutated and have a genetic variation from their origins. Therefore, we suggest the whole-genome sequencing of strains found throughout the country for a better understanding of the viral evolution and identify strains with distinctive mutational changes.

## 5. Conclusions

The surprisingly increased rate of SARS-CoV-2 transmission has forced numerous countries to impose a total lockdown due to an unusual occurrence. Therefore, there is an immediate need to tackle COVID-19, so we want rapid and practical measures. In silico techniques are based on analyzing biological data and using refined predictions and calculations to create a scientific database. In this research, we employed in silico computational tools to analyze SARS-CoV-2 genomes worldwide to detect vital variations in structural proteins and dynamic changes in all SARS-CoV-2 proteins, mainly spike proteins, produced due to a large number of mutations.

Our analysis revealed substantial differences in functionality, immunological, physicochemical, and structural variations in the SARS-CoV-2 isolates. However, the real impact of these SNPs can only be determined by further experiments. Our results can aid in vivo and in vitro experiments in the future.

Current developments in immunological bioinformatics areas have resulted in different servers and tools that can save the cost and time of traditional vaccine development. However, suitable antigen candidates remain a hurdle for researchers. To design a multiple epitope vaccine, the antigenic epitope prediction of a relevant protein by immunoinformatic methods is very helpful. By using in silico cloning, we will acquire a harmless SARS-CoV-2 vaccine that could trigger immune responses: cellular, innate, and humoral. Although the production and manufacture of vaccines are expensive and takes more time, immunoinformatic approaches can decrease this load. Nowadays, researchers are finding different methods to develop multi-epitope subunit vaccines. With the development of computational tools, epitope prediction for antibodies has become more meaningful.

## Figures and Tables

**Figure 1 vaccines-11-00736-f001:**
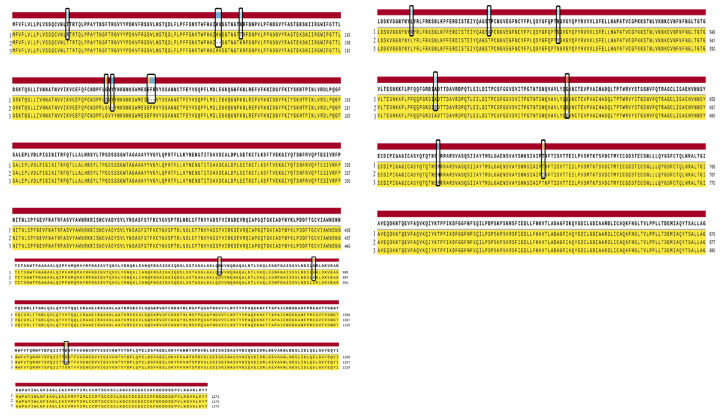
Schematic view of the contigs of the surface glycoprotein of (1) the Alpha variant (B.1.1.7) (UDQ41838.1), (2) the Delta variant (B.1.617.21) (UDU36746.1) of the SARS-CoV-2 with (3) the reference strain (Wuhan) (YP_009724390.1).

**Figure 2 vaccines-11-00736-f002:**
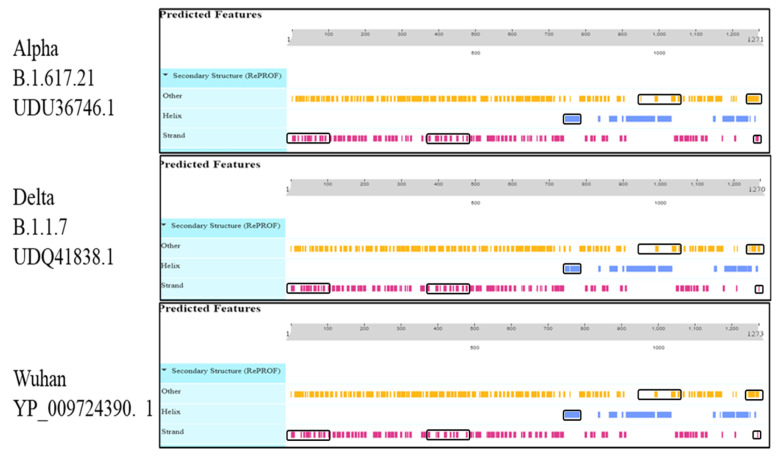
Viewer layout of the predicted features of the protein structural and functional features of the surface glycoproteins.

**Figure 3 vaccines-11-00736-f003:**
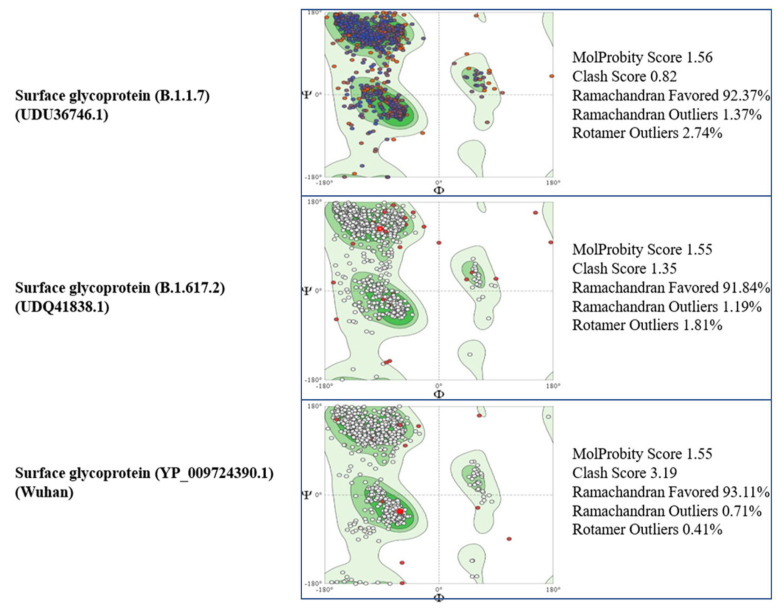
Comparison of the Ramachandran plots and the MolProbity results of the surface glycoproteins.

**Figure 4 vaccines-11-00736-f004:**
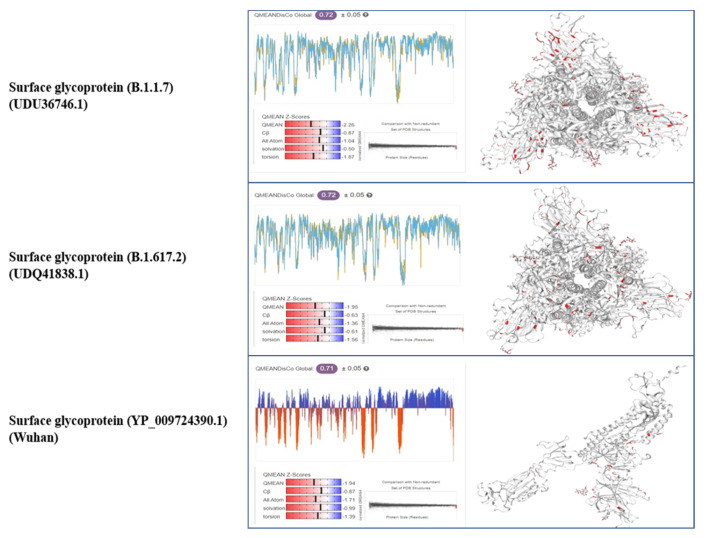
Comparison of the quality estimate and Cryo-EM structure of PCoV_GX of the spike glycoproteins.

**Figure 5 vaccines-11-00736-f005:**
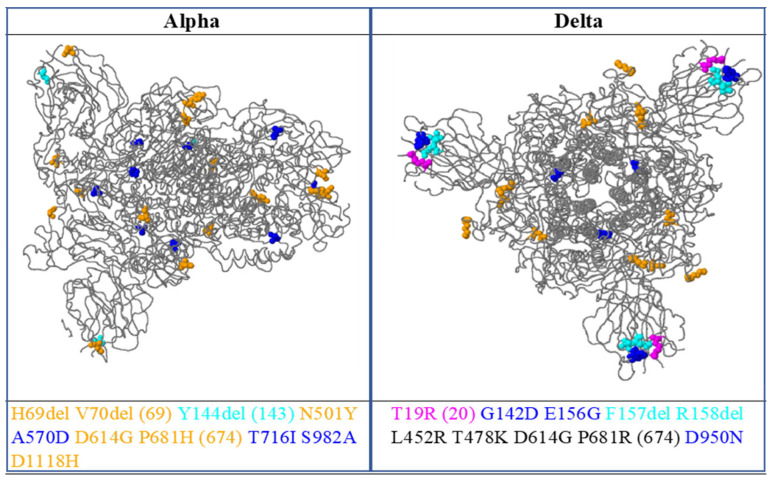
D structural visualization of the spike glycoprotein with amino acid changes, as shown as colored balls.

**Table 1 vaccines-11-00736-t001:** Comparison of the secondary structures of the proteins of SARS-CoV strains B.1.617.21 and B.1.1.7 with a reference strain of SARS-CoV-2.

Proteins	Variants	Accession ID	Alpha Helix (Hh)	Extended Strand (Ee)	Beta Turn (Tt)	Random Coil (Cc)
Envelope protein	B.1.617.21	UDU36748.1	33 is 44%	20 is 26.67%	7 is 9.33%	15 is 20%
	B.1.1.7	UDQ41840.1	33 is 44%	20 is 26.67%	7 is 9.33%	15 is 20%
	Wuhan	YP_009724392. 1	33 is 44%	20 is 26.67%	7 is 9.33%	15 is 20%
Membrane	B.1.617.21	UDU36749.1	81 is 36.49%	48 is 21.62%	13 is 5.86%	80 is 3 6.04%
	B.1.1.7	UDQ41841.1	77 is 34.68%	47 is 21.17%	15 is 6.76%	83 is 37.39%
	Wuhan	YP_009724393. 1	77 is 34.68%	47 is 21.17%	15 is 6.76%	83 is 37.39%
Nucleocapsid	B.1.617.21	UDU36754.1	87 is 20.76%	70 is 16.71%	32 is 7.64%	230 is 54.89%
	B.1.1.7	UDQ41846.1	73 is 17.42%	75 is 17.9%	29 is 6.92%	242 is 57.76%
	Wuhan	YP_009724397. 2	89 is 21.24%	70 is 16.71%	29 is 6.92%	231 is 55.13%
ORF10 Protein	B.1.617.21	UDU36755.1	11 is 28.95%	14 is 36.84%	2 is 5.26%	11 is 28.95%
	B.1.1.7	UDQ41847.1	11 is 28.95%	14 is 36.84%	2 is 5.26%	11 is 28.95%
	Wuhan	YP_009725255. 1	11 is 28.95%	12 is 31.58%	2 is 5.26%	13 is 34.21%
ORF1a	B.1.617.21	UDU36745.1	1773 is 40.25%	911 is 20.68%	353 is 8.01%	1368 is 31.06%
	B.1.1.7	UDQ41837.1	1773 is 40.25%	895 is 20.33%	351 is 7.97%	1383 is 31.42%
	Wuhan	YP_009725295.1	1776 is 40.32%	909 is 20.64%	351 is7. 97%	1369 is 31.08%
ORF1ab Polyprotein	B.1.617.21	UDU36744.1	1937 is 38.75%	1106 is 22.12%	493 is 9.86%	1463 is 29.27%
	B.1.1.7	UDQ41836.1	1944 is 38.91	1106 is 22.14%	497 is 9.95%	1449 is 29%
	Wuhan	YP_009724389.1	1908 is 40.39%	949 is 20.09%	345 is 7.30%	1522 is 32.22%
ORF3a Protein	B.1.617.21	UDU36747.1	72 is 26.18%	80 is 29.09%	28 is 10.18%	95 is 34.55%
	B.1.1.7	UDQ41839.1	72 is 26.18%	82 is 29.82%	28 is 10.18%	93 is 33.82%
	Wuhan	YP_009724391.1	72 is 26.18%	82 is 29.82%	28 is 10.18%	93 is 33.82%
ORF6 Protein	B.1.617.21	UDU36750.1	43 is 70.49%	6 is 9.84%	5 is 8.2%	7 is 11.48%
	B.1.1.7	UDQ41842.1	43 is 70.49%	6 is 9.84%	5 is 8.2%	7 is 11.48%
	Wuhan	YP_009724394.1	43 is 70.49%	6 is 9.84%	5 is 8.2%	7 is 11.48%
ORF7a Protein	B.1.617.21	UDU36751.1	53 is 43.8%	24 is 19.83%	11 is 9.09%	33 is 27.27%
	B.1.1.7	UDQ41843.1	52 is 42.98%	23 is 19.01%	12 is 9.92%	34 is 28.1%
	Wuhan	YP_009724395.1	52 is 42.98%	23 is 19.01%	12 is 9.92%	34 is 28.1%
ORF7b	B.1.617.21	UDU36752.1	32 is 74.42%	1 is 2.33%	1 is 2.33%	9 is 20.93%
	B.1.1.7	UDQ41844.1	32 is 74.42%	1 is 2.33%	1 is 2.33%	9 is 20.93%
	Wuhan	YP_009725318.1	32 is 74.42%	1 is 2.33%	1 is 2.33%	9 is 20.93%
ORF8 protein	B.1.617.21	UDU36753.1	24 is 19.83%	43 is 35.54%	6 is 4.96%	48 is 39.67%
	-	-	-	-	-	-
	Wuhan	YP_009724396.1	24 is 19.83%	43 is 35.54%	6 is 4.96%	48 is 39.67%
Surface Glycoprotein	B.1.617.21	UDU36746.1	381 is 29.98%	271 is 21.32%	40 is 3.15%	579 is 45.55%
	B.1.1.7	UDQ41838.1	377 is 29.69%	284 is 22.36%	45 is 3.54%	564 is 44.41%
	Wuhan	YP_009724390.1	364 is 28.59%	296 is 23.25%	43 is 3.38%	570 is 44.78%

## Data Availability

Not applicable.

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
