# Peer review of "In Silico Analysis of SARS-CoV-2 Spike Proteins of Different Field Variants"

_vaccines, 2023, doi:10.3390/vaccines11040736_

Round 1

Reviewer 1 Report

The manuscript is well-written and interesting. There are just a few minor comments:

1. Please correct SARs-CoV-2 to SARS-CoV-2 (few places)

2. Since the manuscript is dedicated to Spike proteins, it is worthen adding some discussion about its Subunits (S1, S2).  

3. Author Contributions, Funding, and Conflicts of Interest are missing.

Author Response

Hi Sir,

1. Corrected SARs-CoV-2 to SARS-CoV-2 (few places)

2. Added in  the discussion about its Subunits (S1, S2) 

3. Author Contributions, Funding, and Conflicts of Interest are completed.

Reviewer 2 Report

In this study, the authors used in silico computational tools to analyze SARS-CoV-2 genomes worldwide

to detect vital variations in structural proteins and dynamic changes in all SARS-CoV-2 proteins.

This study is interesting and useful to understand the variability of SARS-CoV-2.

However, it is necessary to integrate some parts to make the study more understandable and usable:

1)      Please describe methods in the appropriate section and explain the figure below reported.

2)   The references are quite old since they referred mainly to articles published in 2020.

3)    The authors should explain what are the potential implications of genomic modifications on vaccine strategies and therapy based on antiviral agents.

4)      In the abstract tha authors emphasized: “the situation in the world is getting worse with the emergence of novel variants”. Worldwide, the number of cases of SARS-COV-2 is reducing in the last weeks. It would be better to rephrase the sentence.

5)   The authors should complete: Author Contributions; Funding; Conflicts of Interest and indicate the approval of a Ethic Comitee.

Author Response

Hi Sir,

1) Now method section is in the appropriate section and explained the figure

2) The references are quite old since they referred mainly to articles published in 2020 because most of the articles are published in 2020. 

3)  Explained the potential implications of genomic modifications in vaccine strategies and therapy based on antiviral agents in the conclusion.

4)  Correction in the abstract 

5)   Completed The Author Contributions; Funding; Conflicts of Interest 

Round 2

Reviewer 2 Report

The revised version of the manuscript satisfies the queries addressed.